# A Deep-Learning Extraction Method for Orchard Visual Navigation Lines

**Jianjun Zhou** [1], **Siyuan Geng** [2], **Quan Qiu** [3,*], **Yang Shao** [1] **and Man Zhang** [4,*]

1    College of Information Engineering, Beijing Institute of Petrochemical Technology, Beijing 102617, China
2    Beijing Electro-Mechanical Engineering Institute, Beijing 100074, China
3    Academy of Artificial Intelligence, Beijing Institute of Petrochemical Technology, Beijing 102617, China
4    Key Laboratory of Smart Agriculture System Integration Research, Ministry of Education,
     China Agricultural University, Beijing 100083, China
*    Correspondence: qiuquan0110@ustc.edu (Q.Q.); cauzm@cau.edu.cn (M.Z.); Tel.: +86-10-8129-3195 (Q.Q.);
     +86-10-6273-7188 (M.Z.)

**Abstract:** Orchard machinery autonomous navigation is helpful for improving the efficiency of fruit production and reducing labor costs. Path planning is one of the core technologies of autonomous navigation for orchard machinery. As normally planted in straight and parallel rows, fruit trees are natural landmarks that can provide suitable cues for orchard intelligent machinery. This paper presents a novel method to realize path planning based on computer vision technologies. We combine deep learning and the least-square (DL-LS) algorithm to carry out a new navigation line extraction algorithm for orchard scenarios. First, a large number of actual orchard images are collected and processed for training the YOLO V3 model. After the training, the mean average precision (MAP) of the model for trunk and tree detection can reach 92.11%. Secondly, the reference point coordinates of the fruit trees are calculated with the coordinates of the bounding box of trunks. Thirdly, the reference lines of fruit trees growing on both sides are fitted by the least-square method and the navigation line for the orchard machinery is determined by the two reference lines. Experimental results show that the trained YOLO V3 network can identify the tree trunk and the fruit tree accurately and that the new navigation line of fruit tree rows can be extracted effectively. The accuracy of orchard centerline extraction is 90.00%.

**Keywords:** autonomous navigation; navigation line extraction; orchard machinery; deep learning; least-square

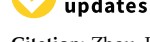



## 1. Introduction

In recent years, the new orchard intelligent machinery has shown great advantages in improving agricultural production efficiency and solving the labor shortage problem. First, such machinery has the ability to avoid direct contact between people and their working environments [1]. For example, there are some toxic or high-temperature scenarios which are not conducive to the human body in some operations. Moreover, the repetitive and monotonous nature of some phases of the orchard fruit production process, such as fruit picking, can be tiring and lead to missed operations or accidents. How to achieve autonomous navigation is one of the hot research topics in the field of intelligent machinery for orchards. With its advantages of wide range of detection information and comprehensive information acquisition, visual navigation has become the most widely used robotic navigation method throughout the world. The key aspect of visual navigation is its accurate and reliable extraction of the navigation baseline through image processing technology [2–4].

For the autonomous navigation problem, research ideas are focused on two aspects: road- or sky-based navigation line generation and crop detection–based fitting of navigation lines. Road- or sky-based navigation methods are highly robust to plant species, shape,

and height and therefore constitute a hot research topic for scholars throughout the world. Crop detection–based navigation methods require accurate identification of crop trunks and are highly robust to complex road environments, thus requiring high adaptability.

Using the features shown in orchard images, He et al. proposed a horizontal projection method to recognize the main trunk area dynamically [5]. The color-difference R-B and two-dimensional Otsu algorithm were employed to segment the trunk from the background. A morphological method was adopted to eliminate noises from tiny branches and fading fallen leaves. Similarly, an optimal path extraction method proposed by Li also adopted color-model and segmentation methods [6]. The least-square and Hough transform methods are the most generally used line fitting methods. Based on the least-square method, both studies fit the reference lines of the fruit trees on both sides. The experimental results showed that the path generation method can provide a theoretical basis and technical support for the walking of a kiwi fruit–picking robot [6].

To achieve a better result, Ali et al. proposed a classification-based tree detection algorithm [7]. Color and texture cues were combined to yield better performance than individual cues could accomplish. Lyu et al. applied the Naive Bayesian classification (Artificial Neural Networks (ANN) and K-nearest neighbor (KNN) in [7]) to detect the boundary between trunk and ground and proposed a method to determine the centerline of orchard rows [8]. The advantage of the Bayesian classification is that it requires a small number of samples and a simple training process. In addition, it can effectively reduce impact from branches, soil, weeds, or tree shadows on the ground. In orchard navigation tests, the steering angle deviations generated by the proposed algorithm were much smaller than those generated from manual decisions. This showed that the orchard navigation method is more stable than a method that determines the centerline extraction manually.

Thus far, most researchers have developed algorithms that take advantage of the ground structures of orchards. These studies use the segmented sky from the tree canopy background and the centroid features of the segmented object as the process variables to guide the unmanned ground vehicle moving in the tree rows [1]. Experiments have shown that these approaches have the potential to guide utility vehicles.

Light detection and ranging (LiDAR) technology is also widely used in orchard navigation. Zhou et al. proposed a method for calculating the center point of the trunk with LiDAR sensory data [9]. LiDARs were used to scan the trunks on both sides of the fruit tree row. Point clusters with approximately circular arc shapes were formed. The central coordinate position and the radius of the trunk could be determined through geometric derivation. As the robot moved, its position and posture were corrected in real time by comparing the detected coordinates of the center point of the trunk with those obtained previously. Blok et al. paid more attention to the robot's self-positioning [3]. This research validated the applicability of two probabilistic localization algorithms that used a single 2D LiDAR scanner for in-row robot navigation in orchards. The first localization algorithm was a particle filter (PF) with a laser beam model, and the second was a Kalman filter (KF) with a line detection algorithm. Experiments were designed to test the navigation accuracy and robustness of the two methods, and the results showed that PF with a laser beam model was preferred over a line-based KF for in-row navigation.

Shalal et al. combined LiDAR and cameras in their research [10,11]. The LiDAR was used to detect edge points to determine the width of trunks and of non-trunk objects. The color and parallel edges of the trunks and non-trunk objects were verified by camera images.

Traditional image processing methods are easily affected by sunlight, canopy occlusion, and weeds. With the development of artificial intelligence, Zhang et al. tried to apply deep learning image processing in orchard management [12]. A multi-class object detection algorithm was proposed on the basis of a region convolutional neural network (R-CNN) model to detect branches, trunks, and apples in the orchard environment. VGG16 and VGG19 (the highest MAP of 82.4%) both achieved higher detection accuracy than Alexnet for the skeleton fitting of branches and trunks [13–15]; this study provided a foundation and possibility for developing a fully automated shake-and-catch apple harvesting system.

According to the above analysis of orchard autonomous navigation research results, the limitations of current orchard navigation are reflected in the following three points: ① In orchards with large tree canopies, it is more difficult to extract the vanishing point, and the application of generating navigation lines based on roads or skies will be limited. ② The use of traditional image processing methods based on tree trunk detection to fit the navigation path is susceptible to light intensity, shadows, and other factors. ③ Using radar data to improve the midpoint of fruit tree trunks provides a method for fruit tree row extraction, and image sensors have the advantage of low cost.

To address the limitations of the existing methods, we provide a DL_LS method that uses a deep learning model to extract the trunks of fruit trees near the ground and calculate the fruit tree reference points, fit the fruit tree row lines through the reference points, and calculate the centerlines through the row lines on both sides. In our method, we employ the YOLO V3 network to detect trunks of fruit trees in contact with the ground area, which can be basically independent of light intensity, shade, and disturbances. Furthermore, we use the detected trunk bounding box to determine the key points or reference points of the tree row, which are the middle points of the bottom lines of the bounding boxes, and then extract the tree row lines by the least-square method in order to improve the accuracy of the tree row line extraction.

Our method consists of four steps: detection of the fruit tree trunks using the deep learning method, determination of the fruit tree reference points, fitting of the fruit tree reference row lines, and generation of the orchard centerlines. The deep convolution neural network, which replaces the traditional feature extraction methods, can automatically detect the target after training with enough sampled learning data. The algorithm of the fruit tree row line fitting is put forward using a least-square algorithm, which can effectively extract the orchard machinery walking route.

## 2. Materials and Methods

The DL-LS algorithm proposed in this study can carry out path planning tasks for autonomous orchard machinery by combining deep learning methods with fruit tree line fitting algorithms. We selected the YOLO V3 network to accurately identify tree trunks with a bounding box, determine key or reference points with the middle points of the bottom lines of the bounding boxes, and fit the tree row reference lines with the least-square algorithm, which can carry out tree row line detection with higher accuracy under different disturbances in orchard scenarios. We collected a large quantity of actual orchard image data. These images were employed to train the YOLO V3 network after the sorting and labeling. Then the coordinates of the bounding box were generated after the tree trucks were detected. The reference point coordinates of the fruit tree can be calculated with these coordinates. The reference lines of the fruit tree rows were fitted by the least-square method. Finally, the centerline of the fruit tree rows was fitted with two reference lines. The principle is shown in Figure 1. This centerline is regarded as the tracking or moving path for the orchard machinery. Figure 2 is a flowchart of the deep learning-based tree/trunk extraction method. In the training stage, images of fruit tree rows in orchards are collected to form a dataset. The dataset is divided into a training set and a test set, and the manual labeling includes two types of tree trunks and fruit trees. The YOLO V3 network is trained using the training set to generate weight files. While testing, the trunk and fruit tree rectangular boxes are generated by the trained network; then fruit tree row reference point coordinates can be obtained by using trunk rectangular box coordinates calculation, and the fruit tree row lines are generated by means of least-squares fitting. Finally, the centerline of the fruit tree rows is obtained using the algorithm.

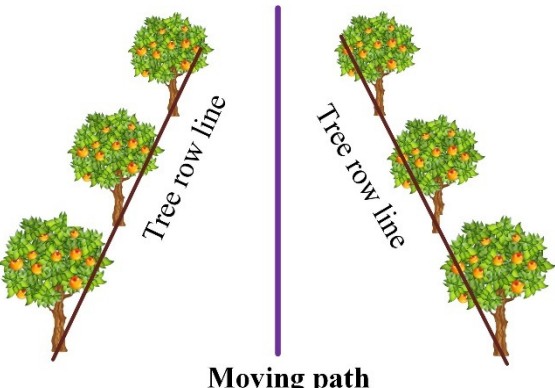

**Figure 1.** Schematic diagram of orchard navigation line extraction.

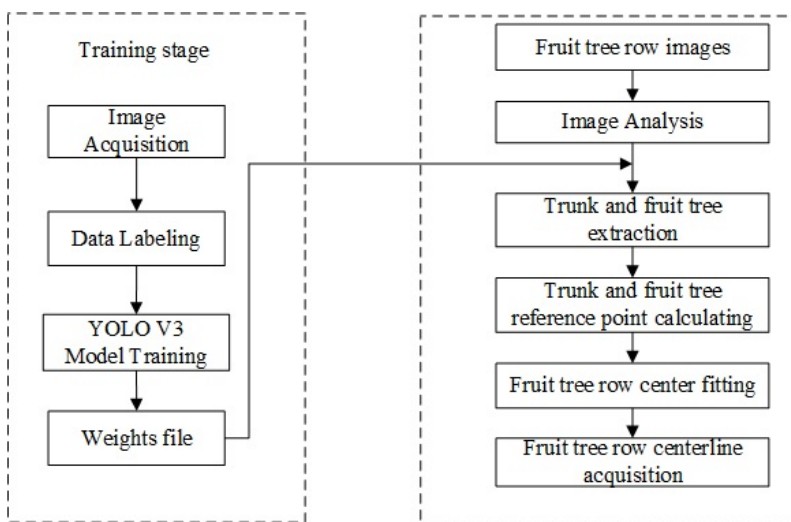

**Figure 2.** Flowchart of the deep-learning extraction method of orchard visual navigation line.

*2.1. Detection of Fruit Tree and Trunk*

Traditional target recognition methods are strongly dependent on specific images and are susceptible to light intensity, shade, etc. In this thesis, the YOLO V3 network is used to identify fruit trees and the trunk of fruit trees in contact with the ground area.

2.1.1. Network Structure of YOLO V3

YOLO V3 uses the residual module to improve the phenomenon of gradient disappearance or gradient explosion, and YOLO V3 borrows the idea of the feature pyramid networks (FPN) algorithm, which has excellent performance for small-target detection. The YOLO v3 network is based on a regression approach to feature extraction, enabling end-to-end object detection. Thus, it is more suitable for field application environments as it can quickly predict and classify targets while ensuring high accuracy.

The backbone network of YOLO V3 is Darknet-53. There are 53 layers of convolutional neural networks. The last layer is the fully-connected layer, and the other 52 layers appear as the layers for feature extraction [16]. The structure is as shown in Figure 3. Moreover, the residual module is widely used in the Darknet-53 network [13]. The gradient will disappear or explode if there are too many layers in the network. The residual module can improve this situation. YOLO V3 adopts the mechanism of multiscale fusion and multiscale prediction. YOLO V3's excellent performance for small-target detection is highly suitable for the task of trunk detection. It uses both the rich detail and location information of the low-level feature map and the rich semantic information of the high-level feature map to improve the detection precision and detect small targets better [17–21].

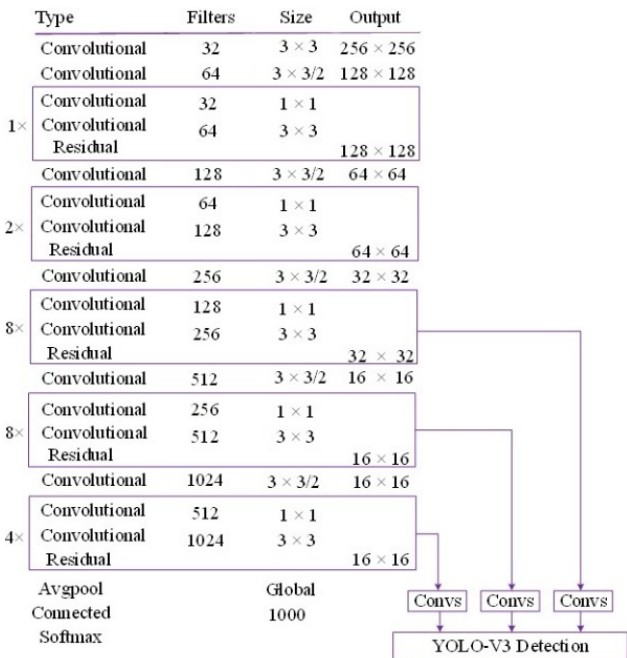

| | Type | Filters | Size | Output |
|---|---|---|---|---|
| | Convolutional | 32 | 3 × 3 | 256 × 256 |
| | Convolutional | 64 | 3 × 3/2 | 128 × 128 |
| 1× | Convolutional | 32 | 1 × 1 | |
| | Convolutional | 64 | 3 × 3 | |
| | Residual | | | 128 × 128 |
| | Convolutional | 128 | 3 × 3/2 | 64 × 64 |
| 2× | Convolutional | 64 | 1 × 1 | |
| | Convolutional | 128 | 3 × 3 | |
| | Residual | | | 64 × 64 |
| | Convolutional | 256 | 3 × 3/2 | 32 × 32 |
| 8× | Convolutional | 128 | 1 × 1 | |
| | Convolutional | 256 | 3 × 3 | |
| | Residual | | | 32 × 32 |
| | Convolutional | 512 | 3 × 3/2 | 16 × 16 |
| 8× | Convolutional | 256 | 1 × 1 | |
| | Convolutional | 512 | 3 × 3 | |
| | Residual | | | 16 × 16 |
| | Convolutional | 1024 | 3 × 3/2 | 16 × 16 |
| 4× | Convolutional | 512 | 1 × 1 | |
| | Convolutional | 1024 | 3 × 3 | |
| | Residual | | | 16 × 16 |
| | Avgpool | | Global | |
| | Connected | | 1000 | |
| | Softmax | | | |

**Figure 3.** The structure of YOLO V3.

### 2.1.2. Image Datasets

The training of deep neural networks requires a great amount of data. The image dataset in this study was acquired from a pear orchard in the Daxing District, Beijing, which contains fruit trees of different ages, including young and adult trees. A large number of images of fruit trees were taken under different angles and illumination. The data collection scenarios are shown in Figure 4. In order to improve the training and prediction speed, the resolution of the input side of the image is uniformly converted to 512 × 512 pixels during image pre-processing. To improve the robustness of the model and suppress overfitting, random perturbations are added to expand the amount of data during training, such as random adjustment of contrast, saturation, brightness, etc. Finally, 971 images are obtained. In each sample image, the position and category of trunks and fruit trees are marked by a rectangle box, and the marked data are saved in a particular format. We chose LabelMe V3.16 installed on Anaconda for image labeling.

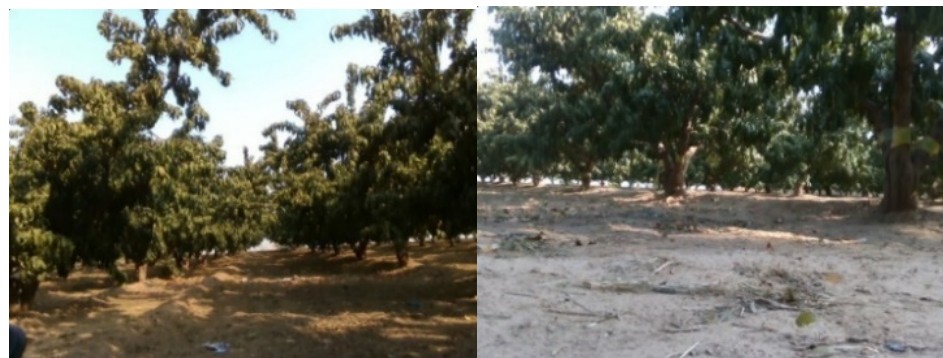

**Figure 4.** Some examples of the image datasets.

### 2.1.3. Model Training

The experiments in this study were conducted on a computer with Intel i7, 64-bit and a GTX 1080Ti GPU. The dataset was split into 70% for training and 30% for testing. In the training and testing processes, the unit of the images was pixel. In the process of model training, there are many hyperparameters that need to be set manually, and the

difference in parameters seriously affects the quality of the model, such as the learning rate and batch size. In our model we set the initial learning rate to 0.001 and the batch size to 8. The learning rate is an important hyperparameter in the deep-learning optimizer which determines the speed of the weight updating. If the learning rate is too high, the training result will exceed the optimal value; if the learning rate is too low, the model will converge too slowly. The batch size depends on the size of the computer memory, and the larger the batch, the better the model training effect. After many parameter adjustments, we trained a model with relatively high accuracy which can accurately identify the trunk and fruit trees in the image. After the training, the loss value curve was drawn, as shown in Figure 5. The line reflects the relationship between the loss value and the number of epochs in the training process. The detection error of YOLO V3 dropped rapidly after the first 10 iterations. And the loss value hardly changed after 50 epochs.

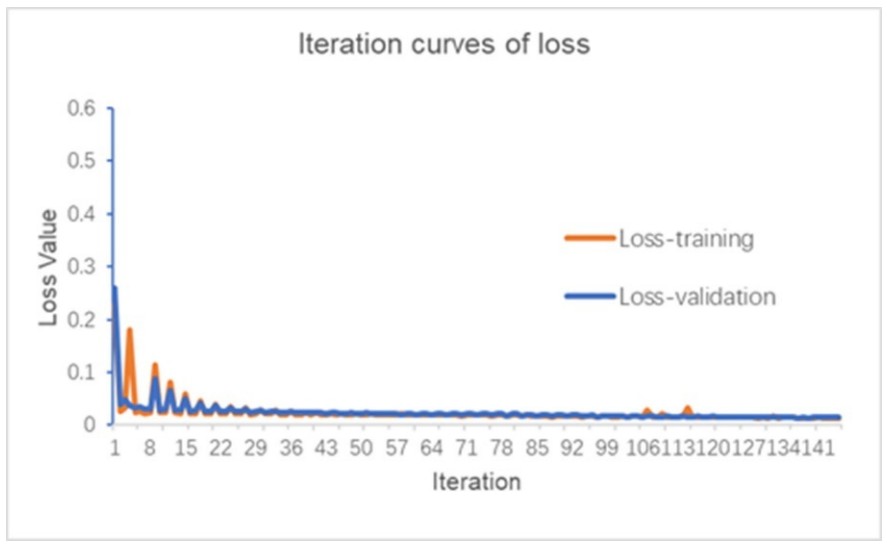

**Figure 5.** Loss curves of the YOLO V3 model.

### 2.2. Path Extraction of Orchard Machinery Navigation

The previous section extracts the information on tree trunk position coordinates from orchard images taken in the row. In this section, the centerline of the fruit tree rows is extracted on the basis of the trunk box coordinates.

### 2.2.1. Reference Point Generation

The coordinate value of the bounding-box border can be read clearly by generating the position information of the trunk, which contains the coordinate value of the points in the upper-left and lower-right corner. The coordinates of the points in the upper-left and lower-right corner are $P_l$ ($x_l$, $y_l$) and $P_r$ ($x_r$, $y_r$), respectively. The reference point of this trunk is ($\frac{x_r - x_l}{2} + x_l$, $y_r$). The algorithm's pseudocodes are shown in Algorithm 1.

**Algorithm 1** Obtain available coordinate points

**Input**: Acquired raw image
          [r c] = size(img)
Imghalfwidth = c/3/2
          A = importdata (txt)
          [m,n] = size (A.data)
1: **for** I = 1:m
2:  **if** textdata including "trunk"
3:     **if** Second data < imghalfwidth
4:        y = The fifth data value in A
5:        x = 0.5 (fourth data value - second data value) + second data value
6:     **else**
7:        y = The fifth data value in A
8:        x = 0.5 (fourth data value - second data value) + second data value
9:     **end**
10:  **end**
11: **end**

### 2.2.2. Line Fitting of the Tree Rows

The reference points of the fruit trees are fitted into the reference lines of the fruit trees on both sides of the row by the least-square method. If there are fewer than three available tree trunks extracted in case of missing fruit trees, we simply connect the nearest two reference points. The process is shown in Algorithm 2.

**Algorithm 2** Obtain the reference lines

**Input**: Sorting the coordinates of the reference points of the left- and right-side fruit trees, respectively
1: **if** the number of points on the left is equal to or greater than 3
2:    least-square method
3: **else if** less than 3 points on the left
4:    Connect two points
5: **end**
6: The right-fitting line is the same as above
7: **if** the number of points on the right is equal to or greater than 3
8:   Fit a straight line using the least-square method
9: **else if** less than 3 points on the left
10: Connect two points line 11
     k = (ycord (1) - ycord (2))/(xcord (1) - xcord (2))
     b = ycord (1)
11: **end**

### 2.2.3. Obtaining the Centerline

The centerline of the previously obtained two reference lines of the fruit tree rows on both sides is the reference line of the orchard machinery, and its detailed principle is shown in Figure 6. We denote point $P_{l1}$ as the farthest reference point on the left reference line in the image. Its corresponding point on the right reference line is $P_{r1}$. We connect the segment $P_{l1}$ $P_{r1}$ and calculate the midpoint $P_{m1}$. Similarly, we denote point $P_{l2}$ as the nearest reference point and connect the segment $P_{l2}$ $P_{r2}$ to determine the point $P_{m2}$. Currently, the straight line passing through $P_{m1}$ and $P_{m2}$ is the reference line for the orchard machinery. The algorithm flow is shown in Algorithm 3.

---

**Algorithm 3** Obtain the centerline

---

**Input**: the left and right reference lines
1:   sort coordinate for the left rectangle label
2:   search the nearest point $P_{l2}$ corresponding Point
$P_{r2}(x_{r2}, y_{r2})$
3:   middle point coordinate $(x_{m2}, y_{m2})$
4:   search the furthest point $P_{l1}$ corresponding Point
$P_{r1} (x_{r1}, y_{r1})$
5:   calculate the coordinates of point $P_{m1} (x_{m1}, y_{m1})$ by points $P_{l1}$ and $P_{r1}$
6:   calculate the coordinates of point $P_{m2} (x_{m2}, y_{m2})$ by points $P_{l2}$ and $P_{r2}$
7:   line connecting points $P_{m1}$ and $P_{m2}$
8:   **end**

---

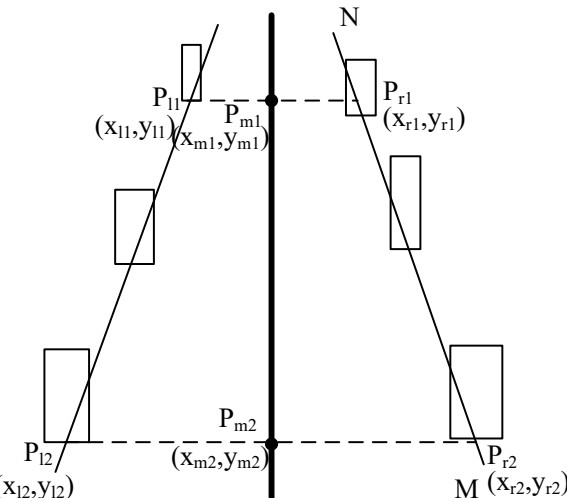

**Figure 6.** Centerline acquisition for orchard machinery.

## 3. Results and Discussion

### 3.1. Tree and Trunk Detection Results

The trained network can identify the tree trunk and fruit tree accurately. The detection accuracy is shown in Table 1. The average precision (AP) of the trees is 92.7%, and the AP of the trunks is 91.51%. The MAP of detection can reach 92.11%, which is not easily affected by sunlight. The trunk of the same fruit tree can be accurately detected under normal sunlight and strong sunlight, as shown in Figure 7. This method has a stronger anti-interference ability compared with traditional methods, especially in the morning and afternoon when the lighting condition changes. Furthermore, weeds easily affect the results of the interference; this is because the color and shape of weeds and leaves are very similar and because weeds occasionally become entangled with the tree trunks. Figure 8 shows the detection result under strong sunlight. The recognition result of the trunks and fruit trees obtained by this network in weed-rich environments shows it to be helpful in alleviating the interference caused by weeds. As shown in Figure 9, the effect of trunk extraction on both sides of the fruit tree rows is excellent under normal sunlight, which is an important basis of this study. Figure 10 shows the result of weak sunlight.

**Table 1.** Detection accuracy.

| Type | AP/% |
|---|---|
| Tree | 92.70 |
| Trunk | 91.51 |

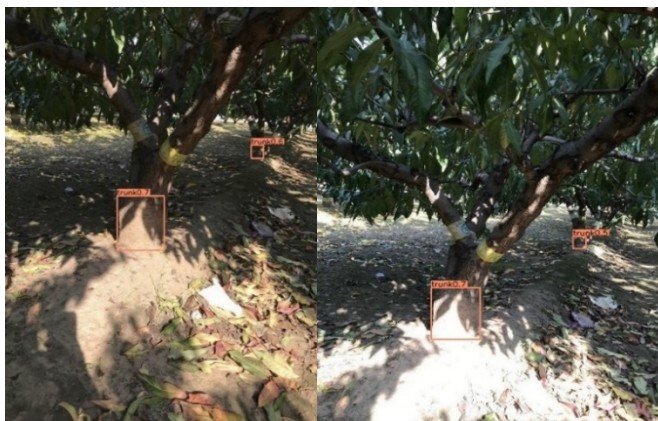

**Figure 7.** Detection results under different sunlight conditions.

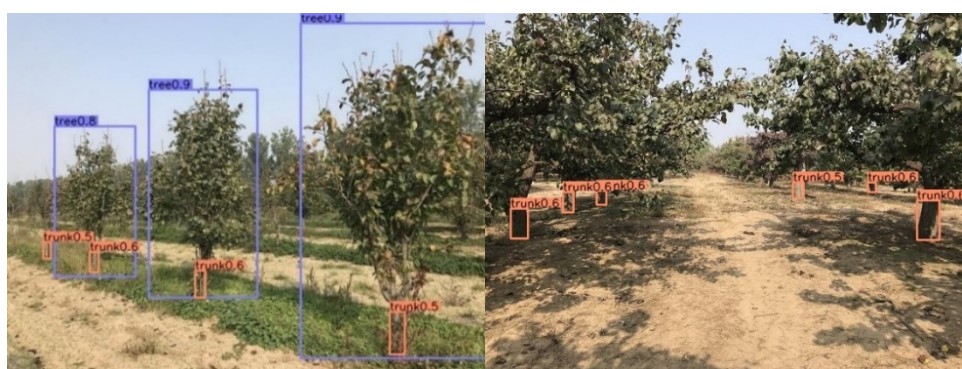

**Figure 8.** Detection results of tree and trunk under strong sunlight.

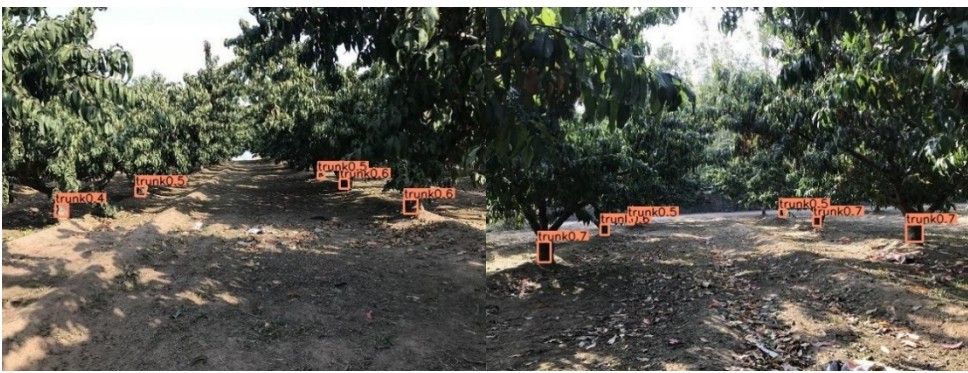

**Figure 9.** Detection results of trunk under normal sunlight.

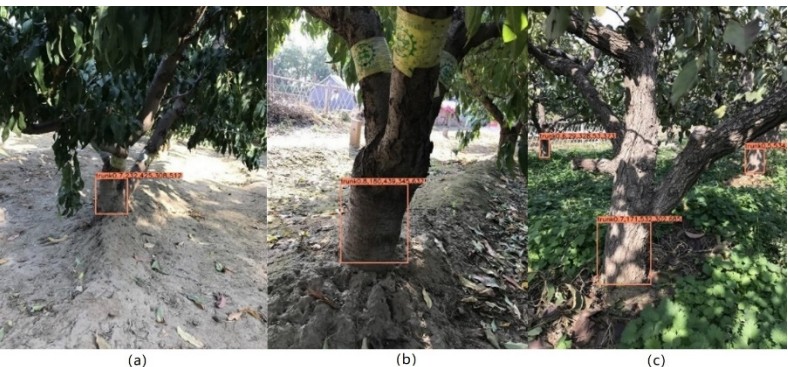

**Figure 10.** Detection results of trunk under weak sunlight.

### 3.2. Results of Reference Point Generation

The accuracy of the algorithm is tested by comparing the coordinate points manually marked with those extracted by our algorithm. The error is calculated by the distance between two pixels. Referring to Figure 10, the error of Figure 10b is larger than errors of the other Figure 10a,c because the height of the tree trunks buried in the soil is irregular, and the error of manually marked points is lower than those obtained in our method. As shown in Table 2, there are five points in three sub-figures. The average error is 1.93 pixels.

**Table 2.** Error analysis of the trunk reference point.

| Sub-Figure | Original Coordinates | Reference Point Coordinates | Manual Marking Coordinates | Error (Pixel) |
|---|---|---|---|---|
| (a) | (232,425) (308,512) | (270,512) | (268,512) | 2.00 |
| (b) | (180,439) (345,632) | (262.5,632) | (260,631) | 2.69 |
|  | (171,532) (302,685) | (236.5,685) | (235,685) | 1.50 |
| (c) | (29,328) (53,373) | (41,373) | (40,372) | 1.41 |
|  | (534,352) (581,411) | (557.5,411) | (558,413) | 2.06 |

### 3.3. Results of Tree-Row Line Fitting

Determining the position of each fruit tree in the image is the basis of orchard mechanical operation, which can obtain the straight line of fruit trees on both row sides. The fitting results of reference lines on both sides of the fruit tree row under different sunlight are shown in Figure 11, including weed environment, strong sunlight, weak sunlight, and normal sunlight. The analysis of the line fitting of fruit tree rows is shown in Table 3.

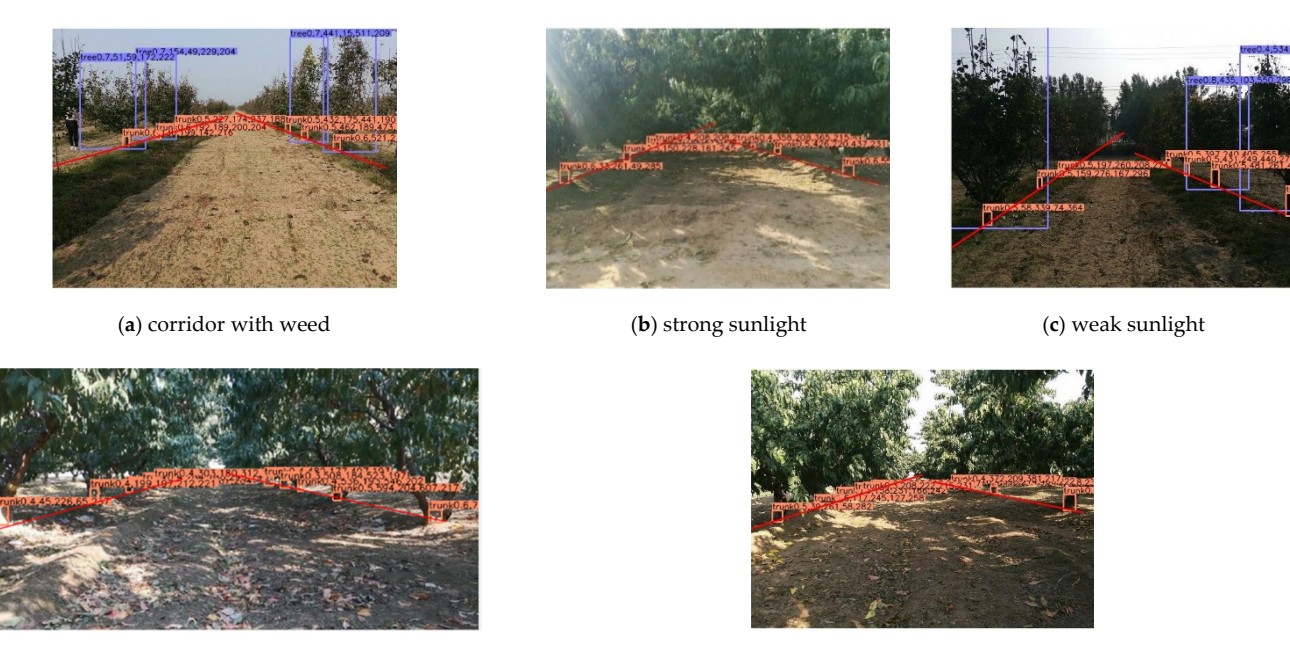

(**a**) corridor with weed     (**b**) strong sunlight     (**c**) weak sunlight

(**d**) normal sunlight on a corridor with leaves     (**e**) normal sunlight

**Figure 11.** Reference line of a fruit tree row under different sunlight.

**Table 3.** Accuracy of line fitting of the fruit tree rows.

|  | Weed Environment | | | Weak Sunlight | | | Strong Sunlight | | | Normal Sunlight | | |
|---|---|---|---|---|---|---|---|---|---|---|---|---|
|  | to the left | to the right | correct | to the left | to the right | correct | to the left | to the right | correct | to the left | to the right | correct |
| Left row line | 1 | 0 | 6 | 1 | 0 | 6 | 0 | 0 | 7 | 1 | 0 | 8 |
| Right row line | 0 | 0 | 7 | 0 | 0 | 7 | 0 | 1 | 6 | 1 | 0 | 8 |
| Total | 1 | 0 | 13 | 1 | 0 | 13 | 0 | 1 | 13 | 2 | 0 | 16 |

Thirty images were selected to test the accuracy of the fruit tree line fit, all of which included both left and right fruit tree rows. The four environments were weeded, low light, high light, and normal light. There were 7 images of the weedy environment, 7 images of the low light environment, 7 images of the high light environment, and 9 images of the normal light environment. When performing this study, the fitted lines should describe the fruit tree rows evenly and accurately, with errors categorized as right-leaning or left-leaning on the left side and right-leaning or left-leaning on the left side. Sixty lines were included in the 30 images, divided into 30 lines on the left and 30 lines on the right side; the specific fruit tree lines fitted are shown in Table 3. In the weed environment, one line on the left side of the tree was fitted to the left, while the other 13 lines were fitted correctly. In the weak-light environment, one line on the left side was fitted to the left, while the other 13 lines were fitted correctly. The right fruit tree row line was fitted to the right in the strong-light environment, while the other 13 were fitted correctly. Under normal light conditions, one line on the left side of the tree was fitted to the left and one line on the right side of the tree was fitted to the right, while the remaining 16 lines were fitted correctly. A total of 55 lines were fitted correctly and 5 lines were fitted inaccurately. The average accuracy of the fruit tree line fits was calculated to be 91.67%.

### 3.4. Results of Centerline Extraction

As shown in Figure 12, the green lines are the centerlines of the orchard machinery. The combination of deep learning and least-square yields a great improvement in efficiency and accuracy compared with traditional methods.

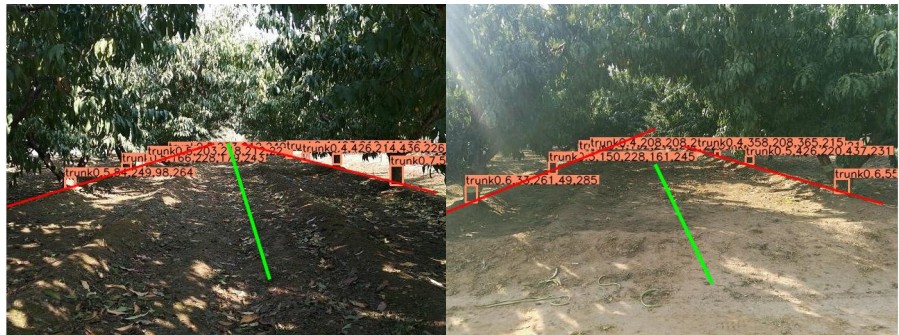

**Figure 12.** Centerlines of fruit rows calculation in the orchard.

In order to evaluate the accuracy of the centerline generation, the benchmark line is selected manually; the difference between the algorithm-generated centerline and the best navigation line is then analyzed. Table 4 shows the fitting results of the centerline in the fruit rows. The accuracy of orchard centerline extraction is 90.00% according to 27 extracted proper centerlines out of 30 images.

**Table 4.** The fitting of the centerline in fruit rows.

|  | Weed Environment | | Weak Sunlight | | Strong Sunlight | | Normal Sunlight | |
|---|---|---|---|---|---|---|---|---|
| type | Little deviation | correct | Little deviation | correct | Little deviation | correct | Little deviation | correct |
| amount | 1 | 6 | 1 | 6 | 0 | 7 | 1 | 8 |

Han, et al. proposed a U-Net network-based approach for visual navigation path recognition in orchards [22]. Table 5 gives a comparative analysis of the maximum and mean value pixel error of the centerline of the fruit tree rows calculated by both U-Net and DL_LS. Under weak light, the maximum pixel error of the centerline is 19 pixels for U-Net and 8 pixels for DL_LS, and the mean value pixel error of the centerline is 11.8 pixels for U-Net and 5.2 pixels for DL_LS; under normal light, the maximum pixel error of the centerline extracted by U-Net is 10 pixels and 5 pixels for DL_LS, and the mean value pixel

error of the centerline extracted by U-Net is 6.5 pixels and 3.4 pixels for DL_LS; under strong light, the maximum pixel error of the centerline extracted by U-Net is 7 pixels and 4 pixels for DL_LS, and the mean value pixel error of the centerline extracted by U-Net is 2.9 pixels and 2.1 pixels for DL_LS. From Table 5, we can infer that our DL_LS can give higher centerline extraction results than those of U-Net.

**Table 5.** Comparison of centerline maximal pixel errors of different methods.

| Method | Weak Sunlight | | Normal Sunlight | | Strong Sunlight | |
|---|---|---|---|---|---|---|
| | Maximum | Mean Value | Maximum | Mean Value | Maximum | Mean Value |
| U-Net [22] | 19 | 11.8 | 10 | 6.5 | 7 | 2.9 |
| DL_LS | 8 | 5.2 | 5 | 3.4 | 4 | 2.1 |

### 3.5. Discussion

Although our method can extract the centerline of two adjacent orchard tree rows with high accuracy, there are still some drawbacks or limitations in our method. First, some of the trunks detected by the deep learning algorithm are side views or parts of the whole trunks, which introduces pixel error while determining the reference points. As a result, the centerline extraction accuracy could be improved if a smart reference point selection strategy is designed. Second, fruit tree trunks of other rows may be captured into the images, so that the extracted feature points are distributed in a zigzag shape, which affects the accurate generation of fruit tree row centerlines. Therefore, a reference or feature point selection or filtering strategy should be proposed to improve our algorithm.

The trained network can identify the tree trunk and fruit tree accurately. The single-target average accuracies for trees and trunks are 92.7% and 91.51% respectively. Trunks and fruit trees are well identified in different sunlight and weed-rich environments. The model has strong robustness, and it takes about 50 milliseconds to process an image, which meets the reliability of the algorithm in real-time mode.

### 4. Conclusions

A centerline extraction algorithm of orchard rows was proposed based on the YOLO V3 network, which can detect fruit trees and trunks in contact with the ground area independent of light intensity, shade, and disturbances. The average detection accuracy of the tree trunks and fruit trees was 92.11% by outputting the coordinate text file of the bounding box at the same time.

With the trunk bounding box, the reference points of the fruit tree trunks were extracted and the least-squares method was applied to fit the fruit tree rows on both sides of the walking routine of the agricultural machinery. According to the experimental results, the centerline of the orchard line was finally fitted. The average accuracy of the fruit tree line extraction was calculated to be 90%.

In the future, our research will consider the fusion of multiple sensors which can acquire richer environmental information and enable automated navigation in complex and changing orchard environments.

**Author Contributions:** Conceptualization, J.Z.; methodology, J.Z and S.G.; validation, J.Z, S.G and Q.Q.; formal analysis, Q.Q., S.G. and J.Z; writing—original draft preparation, J.Z and S.G.; writing—review and editing, J.Z., M.Z. and Q.Q.; visualization, Y.S., S.G. and J.Z.; supervision, M.Z.; funding acquisition, Q.Q. All authors have read and agreed to the published version of the manuscript.

**Funding:** This study was supported by National Natural Science Foundation of China (No.61973040, No. 31101088).

**Institutional Review Board Statement:** Not applicable.

**Data Availability Statement:** The data that support the findings of this study are available from the first author and the second author, upon reasonable request.

**Acknowledgments:** The authors would like to thank Senliu Chen, for his instructive suggestions and kind helps in experiment implementation and paper writing.

**Conflicts of Interest:** The authors declare no conflict of interest.

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
