# Peer review of "A Deep-Learning Extraction Method for Orchard Visual Navigation Lines"

_agriculture, doi:10.3390/agriculture12101650_

Round 1

Reviewer 1 Report

The study is interesting which is proposing a new method for orchard industry. There are some concerns required to be solved and comments are divided into two parts as follows:

Major comments

- It is suggested to prepare a flowchart from the complete procedure of the algorithm.

- It needs more modifications in the paper configuration as follows:

Please fill up the sections ‘’2.1. Detection of fruit tree and trunk’’ and ‘’ 2.2. Path extraction of orchard machinery navigation’’ it remains empty, please explain what readers are going to read shortly.

Change the title ‘’Results’’ to ‘’ Results and discussion’’.

Rename title ‘’ 4. Discussion and Conclusions’’ to ‘’ Conclusion’’. And move some of the discussing sentences to ‘’result and discussion’’ part and prepare a well-written discussion.

Please explain shortly, the ‘’darknet-53 network’’ in page 4 line 4 and reference it.

-How authors assess the reliability of the algorithm in real-time mode?

-what was the size of the input matrix for the algorithm? And it would be interesting to mention the time consumption for the data training.

-It would be interesting to use statistical analysis such as ANOVA or z score or other techniques to see the computed features from the proposed algorithm are significant. Or prepare a statistical analysis that shows significant changes between accuracy results before averaging. It could be a prof of results if it is applicable.

-Please explain the limitations of the proposed algorithm.

-please mention the safety modes in case of error? Doe it may damage if loose the line?

Minor comments

1-In abstract mentioned ‘’reach 92.11%.’’ I am not sure if authors meant 92.11% is the (average) accuracy, specificity, sensitivity or other statistical metrics. Please clarify it.

2- please remove the unnecessary ‘’the’’. Please consider it.

3-There are several abbreviations which are not introduced before that cause confusion for readers, please consider it. For example, FPN, CNN, OTSU and etc.

Author Response

The authors wish to thank the reviewers and the editors for their time and valuable remarks. These efforts and thoughtful comments have led to many improvements in our work. These remarks have been taken under consideration and implemented in this response and in the revision of the paper. Detailed responses to the reviewers and editors are given below.

Major comments

Point 1: It is suggested to prepare a flowchart from the complete procedure of the algorithm.

Response to point 1: We add a flowchart(Figure 2) of the visual navigation line extraction algorithm based on deep learning. Other figure numbers have been modified accordingly. Fig.2 is in line 154, and several sentences are added to explain Fig.2 in Line 142 to 150 as follows:

Fig.2 is a flowchart of the deep learning extraction method. In the training stage, images of fruit tree rows in orchards are collected to form a dataset. The dataset is divided into a training set and a test set, and the manual labeling includes two types of tree trunks and fruit trees. YOLO V3 network is trained using the training set to generate weight files. While testing, the trunk and fruit tree rectangular boxes are generated by the trained network, then fruit tree row reference point coordinates can be obtained by using trunk rectangular box coordinates calculation, and fruit tree row lines are generated by using least squares fitting, and then the center line of the fruit tree rows are obtained by using the algorithm.

Figure 2. Flowchart of deep learning extraction method of orchard visual navigation line

Point 2: It needs more modifications in the paper configuration as follows:

Please fill up the sections ‘’2.1. Detection of fruit tree and trunk’’ and ‘’ 2.2. Path extraction of orchard machinery navigation’’ it remains empty, please explain what readers are going to read shortly.

Response to point 2:We add two paragraphs in line156 to 158 and line 214 to 216 in the revision of the paper to explain what readers are going to read shortly as follows:    

line 156 to 158

“Traditional target recognition methods are strongly dependent on specific images and are susceptible to light intensity, shade, etc. In this paper, YOLO V3 network is used to identify fruit trees and the trunk of fruit trees in contact with the ground part.”

line 214 to 216

“The previous section extracts the information on tree trunk position coordinates from the orchard image taken in the row. In this section, the centerline of the fruit tree rows is extracted based on the trunk box coordinates.”

Point 3: Change the title ‘’Results’’ to ‘’ Results and discussion’’.

Rename title ‘’ 4. Discussion and Conclusions’’ to ‘’ Conclusion’’. And move some of the discussing sentences to ‘’result and discussion’’ part and prepare a well-written discussion.

Response to point 3: Thank you for your advice. We rename the title ‘’3. Result’’ with ‘’3. Result and discussion’’. We rename the title ‘’4. Discussion and Conclusions’’ with ‘’Conclusions’’. 

We add a subsection “3.5 Discussion” and rewrite “4. Conclusions” in line 330 to 358 as follows:

“3.5 Discussion

Although our method can extract the center line of two adjacent orchard tree rows with high accuracy, there are still some drawbacks or limitations in our method. First, some trunks detected by the deep learning algorithm are side-view or parts of the whole trunks, which introduces in pixel error while determining the reference points. As a result, the center line extraction accuracy could be improved if a smart reference point selection strategy is designed. Second, fruit tree trunks of other rows may be captured into the images, so that the extracted feature points are distributed in a zigzag shape, which affects the accurate generation of fruit tree row centerlines. Therefore, a reference or feature point selection or filtering strategy should be proposed to improve our algorithm.

  1. Conclusions

(1) A center line extraction algorithm of orchard rows is proposed based on YOLO V3 network, which can detect fruit trees and trunks in contact with the ground parts independent of light intensity, shade, and disturbances. The average detection accuracy of tree trunks and fruit trees is 92.11% by outputting the coordinate text file of the bounding box at the same time.

(2) With the trunk bounding box, reference points of the fruit tree trunks are extracted and least squares method is applied to fit fruit tree rows on both sides of the marching routine of the agricultural machinery. According to experimental results, the center line of the orchard line was finally fitted. The average accuracy of the fruit tree line extraction is calculated to be 90%.

In future, our research will consider the fusion of multiple sensors which can acquire richer environmental information and enable automated navigation in complex and changing orchard environments.“

Point 4: Please explain shortly, the ‘’darknet-53 network’’ in page 4 line 4 and reference it.

Response to point 4: We modify several sentences to explain ‘’darknet-53 network’’ line in 166 to 168, and added cite Reference in line 411 as follows:

“The backbone network of YOLO V3 is Darknet-53. There are 53 layers of convolutional layer. The last layer is the fully-connected layer, and the other 52 layers appear as the layers for feature extraction[21].”

  1. Luo Z, Yu H, Zhang Y. Pine Cone Detection Using Boundary Equilibrium Generative Adversarial Networks and Improved YOLOv3 Model[J]. Sensors, 2020, 20(16):4430.

Point 5: How authors assess the reliability of the algorithm in real-time mode?

Response point 5: We add a paragrapy in line 340 to 344 to assess the reliability of the algorithm in real-time mode as follows:

“The trained network can identify the tree trunk and fruit tree accurately. The single-target average accuracies for trees and trunks are 92.7% and 91.51% respectively. Trunks and fruit trees are well identified in different sunlight and weed-rich environments. The model has strong robustness, and it takes about 50 milliseconds to process an image, which meets the reliability of the algorithm in real-time mode.”

Point 6: what was the size of the input matrix for the algorithm? And it would be interesting to mention the time consumption for the data training.

Response to point 6: The size of the input image matrix for the algorithm is 512×512 pixels. We add the sentence “the resolution of the input side of the image is uniformly converted to 512×512 pixels during image pre-processing.” in line 184.

Point 7: It would be interesting to use statistical analysis such as ANOVA or z score or other techniques to see the computed features from the proposed algorithm are significant. Or prepare a statistical analysis that shows significant changes between accuracy results before averaging. It could be a prof of results if it is applicable.

Response to point 7: We added a part to compare similar work with our proposed work in line 315 to 329 as follows:

“ Table 5 . Comparison of centerline maximal pixel errors of different methods

Method

Weak sunlight

Normal sunlight

Strong sunlight

Maximum

mean value

Maximum

mean value

Maximum

mean value

U-Net [22]

19

11.8

10

6.5

7

2.9

DL_LS

8

5.2

5

3.4

4

2.1

Han, Z.H. proposed a U-Net network-based approach for visual navigation path recognition in orchards[22]. Table 4 gives a comparative analysis of the maximum and mean value pixel error of the centerline of the fruit tree rows calculated by both U-Net and DL_LS. Under weak light, the maximum pixel error of the centerline is 19 pixels for U-Net and 8 pixels for DL_LS, and the mean value pixel error of the centerline is 11.8 pixels for U-Net and 5.2 pixels for DL_LS; under normal light, the maximum pixel error of the centerline extracted by U-Net is 10 pixels and 5 pixels for DL_LS, and the mean value pixel error of the centerline extracted by U-Net is 6.5 pixels and 3.4 pixels for DL_LS; under strong light, the maximum pixel error of the centerline extracted by U-Net is 7 pixels and 4 pixels for DL_LS, and the mean value pixel error of the centerline extracted by U-Net is 2.9 pixels and 2.1 pixels for DL_LS. From Table 4, we can infer that our DL_LS can give higher centerline extraction results than those of U-Net.”

Point 8: Please explain the limitations of the proposed algorithm.

Response to Point 8: We have the limitations of the proposed algorithm in revision discussions in line 330 to 339 as follows:

”3.5 Discussion

Although our method can extract the center line of two adjacent orchard tree rows with high accuracy, there are still some drawbacks or limitations in our method. First, some trunks detected by the deep learning algorithm are side-view or parts of the whole trunks, which introduces in pixel error while determining the reference points. As a result, the center line extraction accuracy could be improved if a smart reference point selection strategy is designed. Second, fruit tree trunks of other rows may be captured into the images, so that the extracted feature points are distributed in a zigzag shape, which affects the accurate generation of fruit tree row centerlines. Therefore, a reference or feature point selection or filtering strategy should be proposed to improve our algorithm.”

Point 9: please mention the safety modes in case of error? Doe it may damage if loose the line?

Response to Point 9: We think safety modes should appear in the commercial version of the orchard machinery, but somehow out of the range of our paper. In the paper, we provide a method for fruit tree inter-row centerline extraction. In automatic navigation safety modes, computer vision algorithms and robot heading sensors should be combined to provide a safe navigation mode. When the vision detected centerline is failed to be captured, other sensor information will be employed to take the navigation task.

Minor comments

Point 10: In abstract mentioned ‘’reach 92.11%.’’ I am not sure if authors meant 92.11% is the (average) accuracy, specificity, sensitivity or other statistical metrics. Please clarify it.

Response to “minor comments” point 10: 92.11% is mean Average Precision.

We modify the expressions in abstract line 20 as follows.

“After training, mean average precision (mAP) of the model for trunk and tree detection can reach 92.11%.”

Point 11:  please remove the unnecessary ‘’the’’. Please consider it.

 Response to “minor commnets” point 11:We carefully read the whole paper and remove all the unnescessary “the”.

Point 12: There are several abbreviations which are not introduced before that cause confusion for readers, please consider it. For example, FPN, CNN, OTSU and etc.

Response to “minor commnets” point 12:All abbreviations have been introduced in paper.

We add regional convolutional neural network (R-CNN) in line 98 ï¼›

We add feature pyramid networks (FPN) in line 161ï¼›

(OTSU): Otsu algorithm is an efficient algorithm for the binarization of images proposed by the Japanese scholar OTSU in 1979. OTSU is not an acronym, so please forgive me if the reviewer cannot find it. 

OTSU has been modified to Otsu in line 54.

Reviewer 2 Report

The manuscript is intriguing, but I believe it could be improved further. Please see my remarks below:

1. Please use additional references in the first paragraph to back up your points.

2. The article "the" is used excessively throughout the manuscript. Please leave these out where they aren't needed.

3. Please formulate research questions and highlight the manuscript's key contributions. Please also ensure that the research questions investigated are addressed in the results section.

4. Kindly discuss the importance of the work carefully. In the present stage, it has been discussed poorly. This should ideally be done in the introductory section.

5. Please explain the motivation behind using the YOLO V3 network.

6. Figure 4 is bit sloppy. Kindly redraw a better figure.

7. Describe similar work that has been completed. Compare similar work to the your proposed work.

8. Please include a block diagram to describe research methodology. I believe that this would enhance the readability of the paper.

9. I recommend that you thoroughly proofread the manuscript (with the assistance of a native speaker or language editing services or any other services that might be available to you) to improve the manuscript's English language quality.

Author Response

The authors wish to thank the reviewers and the editors for their time and valuable remarks. These efforts and thoughtful comments have led to many improvements in our work. These remarks have been taken under consideration and implemented in this response and in the revision of the paper. Detailed responses to the reviewers and editors are given below.

Point 1: Please use additional references in the first paragraph to back up your points.

Response to Point 1: The authors agree that. We add several sentences in the first paragraph to back up our point. Listed in line 38 to 43 as follows:

“How to achieve autonomous navigation is one of the hot research topics in the field of intelligent machinery for orchards. With its advantages of a wide range of detection information and complete information acquisition, visual navigation technology has become the most used robot navigation method at home and abroad. The key aspect is the accurate and reliable extraction of the navigation baseline through image processing technology [2-3, 20].”

Point 2. The article "the" is used excessively throughout the manuscript. Please leave these out where they aren't needed.

Response to point 2: We carefully read the whole paper and remove all the unnescessary “the”.

Point 3: Please formulate research questions and highlight the manuscript's key contributions. Please also ensure that the research questions investigated are addressed in the results section.

Response to Point 3: We formulate research questions and highlight the manuscript's key contributions.  The research questions investigated are addressed in the result section, listed in line 103 to line 114 as follows:

“According to the above analysis of orchard autonomous navigation research results, the limitations of current orchard navigation are reflected in the following three points:â‘  In orchards with large tree canopies, it is more difficult to extract the vanishing point, and the application of generating navigation lines based on roads or skies will be limited. â‘¡The use of traditional image processing methods based on tree trunk detection to fit the navigation path is susceptible to light intensity, shadows, and other factors.  â‘¢Using radar data to improve the midpoint of fruit tree trunks provides a method for fruit tree row extraction, and image sensors have the advantage of low cost.

To address the limitations of existing methods, we provide a DL_LF method that uses a deep learning method to extract the trunks of fruit trees near the ground and calculate fruit tree reference points, fit the fruit tree row lines through the fruit tree reference points, and calculate the fruit tree row centerlines through the fruit tree row lines on both sides.”

Point 4. Kindly discuss the importance of the work carefully. In the present stage, it has been discussed poorly. This should ideally be done in the introductory section.

Response to Point 4: We discuss the importance of the work in the introductory section in line 103 to 110 as follows: 

“According to the above analysis of orchard autonomous navigation research results, the limitations of current orchard navigation are reflected in the following three points:â‘  In orchards with large tree canopies, it is more difficult to extract the vanishing point, and the application of generating navigation lines based on roads or skies will be limited. â‘¡The use of traditional image processing methods based on tree trunk detection to fit the navigation path is susceptible to light intensity, shadows, and other factors.  â‘¢Using radar data to improve the midpoint of fruit tree trunks provides a method for fruit tree row extraction, and image sensors have the advantage of low cost.”

Point 5: Please explain the motivation behind using the YOLO V3 network.

Response to Point 5: In order to explain the motivation of using YOLO V3 network, we add several sentences in line 160 to 165 as follows:

”YOLO V3 uses the residual module to improve the phenomenon of gradient disappearance or gradient explosion, and YOLO V3 borrows the idea of the feature pyramid networks (FPN )algorithm, which has excellent performance for small target detection. The YOLO v3 network is based on a regression approach to feature extraction, enabling end-to-end object detection. The algorithm is more suitable for field application environments as it can quickly predict and classify targets while ensuring high accuracy. ”

Point 6. Figure 4 is bit sloppy. Kindly redraw a better figure.

Response to Point 6: Figure 4 has been redrawn line in 212 . (Now it is Figure 5 in the paper after adding a block diagram to describe the research methodology )

Point 7. Describe similar work that has been completed. Compare similar work to your proposed work.

Response to Point 7: We add a part of comparison similar work to our proposed work, listed in line 310 to 329 as follows:

“In order to evaluate the accuracy of the centerline generation, the benchmark line is selected manually; the difference between the algorithm-generated centerline and the best navigation line is then analyzed. Table 4 shows the fitting results of the centerline in fruit rows. The accuracy of orchard centerline extraction is 90.00% according to 27 extracted proper centerlines out of 30 images.

Table 4. The fitting of the centerline in fruit rows.

Weed environment

Weak sunlight

Strong sunlight

Normal sunlight

type

Little deviation

correct

Little deviation

correct

Little deviation

correct

Little deviation

correct

amount

1

6

1

6

0

7

1

8

 Table 5 . Comparison of centerline maximal pixel errors of different methods

Method

Weak sunlight

Normal sunlight

Strong sunlight

Maximum

mean value

Maximum

mean value

Maximum

mean value

U-Net [22]

19

11.8

10

6.5

7

2.9

DL_LS

8

5.2

5

3.4

4

2.1

Han, Z.H. proposed a U-Net network-based approach for visual navigation path recognition in orchards[22]. Table 4 gives a comparative analysis of the maximum and mean value pixel error of the centerline of the fruit tree rows calculated by both U-Net and DL_LS. Under weak light, the maximum pixel error of the centerline is 19 pixels for U-Net and 8 pixels for DL_LS, and the mean value pixel error of the centerline is 11.8 pixels for U-Net and 5.2 pixels for DL_LS; under normal light, the maximum pixel error of the centerline extracted by U-Net is 10 pixels and 5 pixels for DL_LS, and the mean value pixel error of the centerline extracted by U-Net is 6.5 pixels and 3.4 pixels for DL_LS; under strong light, the maximum pixel error of the centerline extracted by U-Net is 7 pixels and 4 pixels for DL_LS, and the mean value pixel error of the centerline extracted by U-Net is 2.9 pixels and 2.1 pixels for DL_LS. From Table 4, we can infer that our DL_LS can give higher centerline extraction results than those of U-Net.”  

Point 8. Please include a block diagram to describe research methodology. I believe that this would enhance the readability of the paper.

Response to Point 8: We add a block diagram to describe research methodology in line142 to line150 as follows:

“Fig.2 is a flowchart of the deep learning extraction method of the orchard visual navigation line. In the pre-completion stage, images of fruit tree rows in orchards are collected as the dataset. The dataset is divided into a training set and a test set, and the manual labeling includes two types of tree trunks and fruit trees, and the VOLO V3 network is trained using the training set to generate weight files. When using the test dataset for analysis, the trunk and fruit tree rectangular boxes are generated by the trained network, the fruit tree row reference point coordinates can be obtained by using the trunk rectangular box coordinates calculation, and the fruit tree row lines are generated by using least squares fitting, and then the center line of the fruit tree rows are obtained by using the algorithm.

Figure 2. Flowchart of deep learning extraction method of orchard visual navigation line“

Point 9. I recommend that you thoroughly proofread the manuscript (with the assistance of a native speaker or language editing services or any other services that might be available to you) to improve the manuscript's English language quality.

Response to Point 9: Thank you for your suggestion. We thoroughly proofread the manuscript (with the assistance of a native speaker and a Professor good at English) to improve the manuscript's English language quality.

Reviewer 3 Report

The manuscript is written with clear understanding of the project addressed. However, there are some concerns that need to be addressed to enhance the quality of the manuscript especially in results and discussion. My specific comments are as follows:

Introduction:

“Finally, the experiment showed that this method was more stable than the manual center line extraction method.” Stable in what way?

“These works use the segmented sky from the tree canopy background, and the centroid features of the segmented object as…” add citation

Based on your objectives, please compare how your study is different from those that have already been published

Methods:

“The DL-LS algorithm proposed in this study can…” explain the algorithm

What type of imaging unit is used in this study? Explain

Elaborate more on data enhancement used

What are the hyperparameters used in this study? Explain

Add data analysis section

Results and discussion:

Explain acronym for AP and mAP

“Thirty images were selected to test the accuracy of the fruit tree line fit, all…” is this for validation dataset?

The findings lack in terms of justification and major findings.

Conclusions:

Separate the discussion and conclusion parts. Or combine results and discussion section

Revise the conclusion. Add major finding of your study

Add recommendation for future studies

General comments:

Please check the reference styles and grammar of the manuscript.

Author Response

The authors wish to thank the reviewers and the editors for their time and valuable remarks. These efforts and thoughtful comments have led to many improvements in our work. These remarks have been taken under consideration and implemented in this response and in the revision of the paper. Detailed responses to the reviewers and editors are given below.

Introduction:

Point 1: “Finally, the experiment showed that this method was more stable than the manual center line extraction method.” Stable in what way?

Response to “Introduction” point 1: In the revised version, line 69 to 72, we replace the original sentence of “Finally, the experiment showed that this method was more stable than the manual center line extraction method.” into “In orchard navigation tests, the steering angle deviations generated by the proposed algorithm was much smaller than manual decisions, which showed that this method was more stable than that of manually determining the center line.”

Point 2:“These works use the segmented sky from the tree canopy background, and the centroid features of the segmented object as…” add citation

Response to “Introduction” point 2: Citation has been added in line 74 to 76 as follows: “These works use the segmented sky from the tree canopy background, and the centroid features of the segmented object as the process variables to guide the unmanned ground vehicle moving in the tree rows[1].” And the added citation lies in line 364 as:

  1. Radcliffe, J.; Cox, J.; Bulanon, D. M. Machine vision for orchard navigation. Computers in Industry 2018, 98, 165-171. DOI: https://doi.org/10.1016/j.compind.2018.03.008

Point 3: Based on your objectives, please compare how your study is different from those that have already been published

Response to “Introduction” point 3: We add a paragraph to explain the differences between our method and others, and listed the good features of our method in line 103 to 120 as follows:

“According to the above analysis of orchard autonomous navigation research results, the limitations of current orchard navigation are reflected in the following three points:â‘  In orchards with large tree canopies, it is more difficult to extract the vanishing point, and the application of generating navigation lines based on roads or skies will be limited. â‘¡The use of traditional image processing methods based on tree trunk detection to fit the navigation path is susceptible to light intensity, shadows and other factors.  â‘¢Using radar data to improve the midpoint of fruit tree trunks provides a method for fruit tree row extraction, and image sensors have the advantage of low cost.

To address the limitations of existing methods, we provide a DL_LS method that uses a deep learning method to extract the trunks of fruit trees near the ground, calculate fruit tree reference points, fit the fruit tree row lines through the fruit tree reference points, and calculate the fruit tree row centerlines through the fruit tree row lines on both sides. In our method, we employ VOLO V3 network to detect trunks of fruit trees in contact with the ground part, which can be basically independent of light intensity, shade, and disturbances. Furthermore, we use the detected trunk bounding box to determine the key points or reference points of the tree row, which are the middle points of the bottom lines of the bounding boxes, and then extract the tree row lines with least square method, in order to improve the tree row line extraction accuracy.”

Methods:

Point 1: “The DL-LS algorithm proposed in this study can…” explain the algorithm

Response to “Methods” Point 1: We add a short paragraph to explain the algorithm in line 130 to 135 after “The DL-LS algorithm proposed in this study can…” as follows:

“The DL-LS algorithm is proposed by combining deep learning methods with fruit tree line fitting algorithms. Here, we select YOLO V3 network to accurately identify tree trunks with bounding box, determine key or reference points with the middle points of the bottom lines of bounding boxes, and then fit the tree row reference lines with least-square algorithm, which can carry out tree row line detection with higher accuracy under different disturbances in orchard scenarios.”

Point 2: What type of imaging unit is used in this study? Explain

Response to “Methods” Point 2: The imaging unit is pixel. We add the explanation in line 196 as  “In the training and testing processes, the unit of images is pixel.”

Point 3: Elaborate more on data enhancement used

Response to “Methods” Point 3: We add several sentences to elaborate data enhancement in line 184 to 188 as follows:

“In order to improve the training and prediction speed, the resolution of the input side of the image is uniformly converted to 512×512 pixels during image pre-processing. To improve the robustness of the model and suppress overfitting, random perturbations are added to expand the amount of data during training, such as random adjustment of contrast, saturation, brightness, etc.”

Point 4 :What are the hyperparameters used in this study? Explain

Response to “Methods” Point 4: We add several sentences to the hyperparameters used in this study in line 197 to 207 as follows:

“In the process of model training, there are many hyperparameters that need to be set manually, and the difference of parameters seriously affects the quality of the model. Such as the learning rate and batch size. In our model we set the initial learning rate to 0.001 and the batch size to 8. The learning rate is an important hyperparameter in the deep learning optimizer, which determines the speed of weight update. If the learning rate is too high, the training result will exceed the optimal value; if the learning rate is too low, the model will converge too slowly. The batch size depends on the size of the computer memory, and the larger the batch, the better the model training effect. After many times of parameter adjustment, we trained a model with relatively high accuracy, which can accurately identify the trunk and fruit trees in the image.”

Point 5 : Add data analysis section

Response to “Methods” Point 5 : We add several sentences to analysis the collected data in “2.1.2. Image dataset”, line 180 to 183 as follows:

“The image dataset was acquired from a pear orchard in Daxing District, Beijing, which contains fruit trees of different ages, including adult and young trees. A large number of images of fruit trees were taken under different angles and illumination.”

Results and discussion:

Point 1: Explain acronym for AP and mAP

Response to “Results and discussion” Point 1: We add the full name of AP in line 244 as “average pecision (AP) ” and the full name of mAP in line 20 as “mean average precision (mAP )”

Point 2: “Thirty images were selected to test the accuracy of the fruit tree line fit, all…” is this for validation dataset?

Response to “Results and discussion” Point 2: Yes, “Thirty images were selected to test the accuracy of the fruit tree line fit, all…” is for validation dataset.

Point 3: The findings lack in terms of justification and major findings.

Response to “Results and discussion” Point 3: We rewrite the “Conclusions” to elaberate the justification and major findings in line 346 to 355 as follows:

“(1) A center line extraction algorithm of orchard rows is proposed based on YOLO V3 network, which can detect fruit trees and trunks in contact with the ground parts independent of light intensity, shade, and disturbances. The average detection accuracy of tree trunks and fruit trees is 92.11% by outputting the coordinate text file of the bounding box at the same time.

(2) With the trunk bounding box, reference points of the fruit tree trunks are extracted and least squares method is applied to fit fruit tree rows on both sides of the marching routine of the agricultural machinery. According to experimental results, the center line of the orchard line was finally fitted. The average accuracy of the fruit tree line extraction is calculated to be 90%.”

Conclusions:

Point 1: Separate the discussion and conclusion parts. Or combine results and discussion section

Revise the conclusion. Add major finding of your study

Add recommendation for future studies

Response to “Conclusions” Point 1: Thank you for your advice. Results and discussion section are combined, and conclusion has been revised in line 330 to 358 as follows:

“3.5 Discussion

Although our method can extract the center line of two adjacent orchard tree rows with high accuracy, there are still some drawbacks or limitations in our method. First, some trunks detected by the deep learning algorithm are side-view or parts of the whole trunks, which introduces in pixel error while determining the reference points. As a result, the center line extraction accuracy could be improved if a smart reference point selection strategy is designed. Second, fruit tree trunks of other rows may be captured into the images, so that the extracted feature points are distributed in a zigzag shape, which affects the accurate generation of fruit tree row centerlines. Therefore, a reference or feature point selection or filtering strategy should be proposed to improve our algorithm.

The trained network can identify the tree trunk and fruit tree accurately. The single-target average accuracies for trees and trunks are 92.7% and 91.51% respectively. Trunks and fruit trees are well identified in different sunlight and weed-rich environments. The model has strong robustness, and it takes about 50 milliseconds to process an image, which meets the reliability of the algorithm in real-time mode.

  1. Conclusions

(1) A center line extraction algorithm of orchard rows is proposed based on YOLO V3 network, which can detect fruit trees and trunks in contact with the ground parts independent of light intensity, shade, and disturbances. The average detection accuracy of tree trunks and fruit trees is 92.11% by outputting the coordinate text file of the bounding box at the same time.

(2) With the trunk bounding box, reference points of the fruit tree trunks are extracted and least squares method is applied to fit fruit tree rows on both sides of the marching routine of the agricultural machinery. According to experimental results, the center line of the orchard line was finally fitted. The average accuracy of the fruit tree line extraction is calculated to be 90%.

In the future, our research will consider the fusion of multiple sensors which can acquire richer environmental information and enable automated navigation in complex and changing orchard environments. “

General comments:

Please check the reference styles and grammar of the manuscript.

 Response to “General comments”: The reference styles and grammar of the manuscript has been checked under the help of native English speakers and professional English teachers.

Round 2

Reviewer 1 Report

My concerns are solved.

Author Response

Thank you very much for your kind review comments! We think all the points are valuable and have been addressed. For the second round of revision, we go through the paper carefully again and check the grammar/expression problems. We believe the second revision is well prepared for publish.

Reviewer 2 Report

Thank you so much for revising the manuscript. The authors have addressed all of my concerns, and the revised version of the manuscript is noticeably better than the original. I would suggest that the authors remove the numbering in the conclusion section.

Author Response

Thank you very much for your kind comments. We remove the numbering in the conclusion section following your suggestion.